# Wet-Spun Polycaprolactone Scaffolds Provide Customizable Anisotropic Viscoelastic Mechanics for Engineered Cardiac Tissues

**DOI:** 10.3390/polym14214571

**Published:** 2022-10-28

**Authors:** Phillip R. Schmitt, Kiera D. Dwyer, Alicia J. Minor, Kareen L. K. Coulombe

**Affiliations:** Center for Biomedical Engineering, School of Engineering, Brown University, Providence, RI 02912, USA

**Keywords:** PCL, polycaprolactone, engineered cardiac tissue, wet spinning, anisotropic, cardiac tissue scaffold

## Abstract

Myocardial infarction is a leading cause of death worldwide and has severe consequences including irreversible damage to the myocardium, which can lead to heart failure. Cardiac tissue engineering aims to re-engineer the infarcted myocardium using tissues made from human-induced pluripotent stem cell-derived cardiomyocytes (hiPSC-CMs) to regenerate heart muscle and restore contractile function via an implantable epicardial patch. The current limitations of this technology include both biomanufacturing challenges in maintaining tissue integrity during implantation and biological challenges in inducing cell alignment, maturation, and coordinated electromechanical function, which, when overcome, may be able to prevent adverse cardiac remodeling through mechanical support in the injured heart to facilitate regeneration. Polymer scaffolds serve to mechanically reinforce both engineered and host tissues. Here, we introduce a novel biodegradable, customizable scaffold composed of wet-spun polycaprolactone (PCL) microfibers to strengthen engineered tissues and provide an anisotropic mechanical environment to promote engineered tissue formation. We developed a wet-spinning process to produce consistent fibers which are then collected on an automated mandrel that precisely controls the angle of intersection of fibers and their spacing to generate mechanically anisotropic scaffolds. Through optimization of the wet-spinning process, we tuned the fiber diameter to 339 ± 31 µm and 105 ± 9 µm and achieved a high degree of fidelity in the fiber structure within the scaffold (fiber angle within 1.8° of prediction). Through degradation and mechanical testing, we demonstrate the ability to maintain scaffold mechanical integrity as well as tune the mechanical environment of the scaffold through structure (Young’s modulus of 120.8 ± 1.90 MPa for 0° scaffolds, 60.34 ± 11.41 MPa for 30° scaffolds, 73.59 ± 3.167 MPa for 60° scaffolds, and 49.31 ± 6.90 MPa for 90° scaffolds), while observing decreased hysteresis in angled vs. parallel scaffolds. Further, we embedded the fibrous PCL scaffolds in a collagen hydrogel mixed with hiPSC-CMs to form engineered cardiac tissue with high cell survival, tissue compaction, and active contractility of the hiPSC-CMs. Through this work, we develop and optimize a versatile biomanufacturing process to generate customizable PCL fibrous scaffolds which can be readily utilized to guide engineered tissue formation and function.

## 1. Introduction

Heart disease continues to be the leading cause of death in the United States and worldwide, killing 655,000 Americans each year [1]. Ischemic damage to the ventricular wall due to myocardial infarction (MI) is characterized by muscle death and noncontractile scar tissue formation, which increases the risk of ventricular wall thinning and dilation. Further, two-thirds of patients suffer MI that progresses to heart failure, for which the current standard of care is a heart transplant [2]. Current clinical therapies for ischemic injury to curb its progression to heart failure aim to reduce risk factors such as high blood pressure, but are unable to restore contractility in ischemic tissue [3]. Cardiac tissue engineering seeks to regenerate the infarcted myocardium to restore healthy contractile function using implantable biomaterials and cellularized constructs. One leading strategy to accomplish this is to re-engineer the myocardium using tissues formed with human-induced pluripotent stem cell-derived cardiomyocytes (hiPSC-CMs) cultured in vitro, and subsequently implanting them as a patch on the epicardium [2,4]. The tissues are typically engineered with a scaffold made from synthetic polymers or bio-derived materials including proteins and decellularized ECM [4,5]. These scaffolds offer mechanical support in vivo and can aid in hiPSC-CM alignment and maturation in vitro [4,6], as well as improve mechanical robustness for implantation. Tissue scaffolds are also used to tune the mechanical properties of the engineered tissue, as it has been shown that CMs function best in mechanical environments similar to the native myocardium [7,8,9,10].

A key feature of myocardial structure is its mechanical anisotropy, due to the aligned orientation of muscle fibers that allow for large reductions in ventricular wall size during contraction to facilitate the efficient pumping of high volumes of blood [2]. Anisotropy is characterized by a difference in stiffness along orthogonal directions of the tissue. Recent studies on scaffold anisotropy in hiPSC-CM cultures have shown that scaffolds with aligned or directional fibers promote significant CM alignment and can even aid in promoting CM maturation in engineered tissues [11,12,13].

Mechanical support is useful not only for supporting tissues grown in vitro, but also as a therapy for the injured heart. The passive mechanical reinforcement of infarcted regions using biomaterials has been shown to promote a chamber-level improvement of parameters, such as ejection fraction, which are adversely impacted by heart failure [14,15]. Furthermore, mechanical anisotropy can play a vital role when incorporated into implantable constructs by decreasing strain at the infarction site through longitudinal reinforcement [2,11]. Thus, a cardiac patch that facilitates anisotropic support not only has the potential to improve engineered tissue function, but also to reinforce and mitigate further damage in the infarcted heart wall. The use of a synthetic scaffold for this purpose rather than a bio-derived scaffold offers distinct advantages in ease of manufacturability, lower cost, and consistency between units.

Polycaprolactone (PCL), a polymer used extensively in bone and skin tissue engineering, has shown promise as a cardiac tissue scaffold material due to its mechanical and biological compatibility with heart tissue, as well having FDA approval for implantable devices [16,17,18]. Additionally, its biodegradability over a period of 2–3 years is an attractive feature that could lead to the better long-term integration of cardiac patches after early mechanical support unloads the heart to promote muscle recovery [16,19,20]. PCL scaffolds are most commonly fabricated via electro-spinning, which uses a high voltage to produce nanoscale fibers [16,21,22]. However, this technique is limited in its ability to precisely control orientation and spacing between individual fibers, causing low porosity, and thus, little room for cells [23]. There is a notable need for a scaffold that not only facilitates engineered tissue formation, but which can also be tuned in a controlled way to provide directional mechanical cues to influence cardiac tissue structure. Fortunately, fiber fabrication via wet-spinning produces larger individual microfibers that can easily be manipulated robotically to achieve the desired, precise scaffold geometries.

Herein, we present a novel scaffold design using wet-spun PCL microfibers with customized diameter and inter-fiber angles that allow for precise manipulation of the mechanical anisotropy of the scaffold. The mechanical properties of single wet-spun PCL fibers as well as scaffolds are reported. Further, the scaffolds are combined with hiPSC-CMs in a collagen hydrogel to form a composite engineered cardiac tissue. The impacts of the scaffold on engineered tissue formation and function are assessed by two-dimensional compaction, immunohistochemical analyses, as well as passive mechanical properties and active contractile force generation. This resulted in mechanically supported, contractile tissues. To our knowledge, this is the first report of wet-spun PCL microfiber use to control scaffold anisotropy for a tissue-engineering application, opening new avenues for scaffold design and the development of engineered tissues.

## 2. Materials and Methods

### 2.1. PCL Scaffold Production

PCL (Mn = 80,000, 20% *w*/*v*, Sigma Aldrich, Saint Louis, MO, USA) was dissolved in acetone (80 mL, Sigma Aldrich, Saint Louis, MO, USA)) at 37 °C. The solutions were placed into a 5 mL syringe with an attached blunt 21-gauge or 30-gauge needle (McMaster Carr) and pumped at a rate of 5 mL/h into a 70% *v*/*v* solution of ethanol in water (Figure 1A). The fibers were allowed to solidify in the ethanol solution and were subsequently collected onto a spool and air-dried for a minimum of 24 h.

The dry fibers were guided onto an Arduino-controlled mandrel capable of translation and rotation, as we previously published (with CAD files) [24]. An Arduino Uno microcontroller attached to two motors allowed the precise positioning of fibers on the mandrel (Figure 1B). Scaffolds were produced with 6 fibers each, spaced at 400 µm, at fiber intersection angles of 0°, 30°, 60°, and 90°. The manufactured scaffolds were captured in a sterile biological safety cabinet using autoclaved custom-designed 3D-printed polycarbonate frames with attached polydimethylsiloxane (PDMS) gaskets (Figure 1C,D). Once the scaffolds were secure within frames, excess fibers were cut and the scaffolds placed in sterile 6-well plates.

### 2.2. Degradation Analysis of PCL Fibers

To assess the loss of mass and mechanical integrity due to PCL degradation in physiological salt conditions, 3 cm sections of fibers (*n* = 5 for each condition and time point, total *n* = 80) were cut and weighed individually on an analytical balance (Mettler Toledo XS205). The fiber sections were then placed into phosphate-buffered saline (PBS; Thermo Fisher Scientific, Waltham, MA, USA) at 37 °C. After a defined period (up to 30 days), the fibers were removed and rinsed with deionized water and allowed to dry. They were weighed again and subjected to tensile testing.

### 2.3. Mechanical Testing of Scaffolds

The tensile properties of single fibers were measured using an Instron 5940 equipped with a 500 N load cell. Briefly, standardized 5 cm sections of fiber were cut and attached to the testing rig using screw side action tensile grips. Extension was performed at a rate of 10 mm/min (50% strain/min). Load, extension, and initial length were recorded.

Stress–strain curves, Young’s modulus, and ultimate tensile strength were calculated using load and extension data from tensile testing to failure, as well as fiber diameter data from the image analysis. Young’s modulus was taken as the slope of a linear fit to the stress–strain relationship and is reported in binned increments of strain (0–5%, 5–10%, 10–15%, and 15–20% elongation). Ultimate tensile strength was determined as the maximum stress reached during the tensile test.

The viscoelastic properties of bare (acellular, dry), whole PCL scaffolds were similarly measured in tension using an Instron 5940 with a 500 N load cell. The standardized scaffold samples were preconditioned for five cycles from 0 to 10% strain at a rate of 1% strain/s. Subsequently, each sample was pulled to 15% strain and returned to 0% strain at three distinct rates: 0.1% strain/second, 1% strain/second, and 10% strain/second. Load, extension, and initial length were recorded. Stress–strain curves were generated by estimating the cross-sectional area as equal to six times the cross-sectional area of a single fiber. Young’s modulus was calculated from 5 to 10% strain, as described previously. The hysteresis area was calculated as the area between the loading and unloading curves for each cyclic test.

### 2.4. Cardiomyocyte Differentiation, Expansion, and Lactate Purification

Cardiomyocytes (CMs) were differentiated from hiPSCs in CDM3 Medium (Thermo Fisher Scientific, Waltham, MA, USA) using a previously established Wnt-signaling activation and inhibition protocol (Figure 1E) [25,26]. Briefly, hiPSCs were cultured in Essential 8 Medium (E8 medium; Thermo Fisher Scientific, Waltham, MA, USA) with 5 µM of ROCK Inhibitor (RI; Fisher) on Geltrex (Thermo Fisher Scientific, Waltham, MA, USA)-coated plates until they reached 90% confluency. On day 0 of differentiation, hiPSCs were treated with 3.5 µM of Chiron 99,021 (Chiron; Tocris, Bristol, United Kingdom) to activate Wnt signaling; then, at day 1 (24 h later), the Chiron was removed. On day 3, the cells were treated with 5 µM IWP2 (Tocris) to inhibit the Wnt signaling pathway. A beating CM phenotype was visible between days 10 and 12, at which point the CMs were harvested using TrypLE Select Enzyme (Thermo Fisher Scientific, Waltham, MA, USA) and replated in RPMI 1640 Medium with B27 supplement (RPMI + B27; Thermo Fisher Scientific, Waltham, MA, USA) on 15 cm tissue culture dishes coated with Geltrex.

For expansion following an established protocol [27], low-density, replated CMs were treated with 2 µM of Chiron until confluence was reached (typically 4–6 days). The CMs were then subjected to four days of Wnt inhibition by treatment in 2 μM of XAV939 (Tocris, Bristol, United Kingdom). For lactate selection to improve CM purity following an established protocol [28], the culture medium was changed to DMEM—glucose + 4 mM sodium lactate (Thermo Fisher Scientific). After four days in lactate medium, the cells were returned to RPMI + B27 medium. The cells were replated once more in RPMI + B27 medium prior to harvest for engineered tissues. The purity of the hiPSC-CMs used for the engineered tissues was >70%, as assessed by a flow cytometry analysis for cardiac troponin T (cTnT).

### 2.5. Scaffold Preparation and Engineered Tissue Formation

Captured PCL scaffolds were soaked in RPMI + B27 medium overnight to allow for protein adsorption for the coating of PCL fibers. Lactate-purified CMs were harvested using TrypLE Select Enzyme and resuspended, along with 5% adult human cardiac fibroblasts (Sigma) in RPMI + B27 medium at a concentration of 30 × 10^6^ cells/mL. An equal volume of 2.4 mg/mL rat tail collagen-1 (Advanced BioMatrix RatCol) was added at a 50%/50% *v*/*v* ratio for a final concentration of 15 × 10^6^ cells/mL and 1.2 mg/mL collagen. This solution was pipetted into captured scaffolds or PDMS molds (controls) and incubated for 30 min at 37 °C for collagen gellation. Once the hydrogel formed, tissue constructs were maintained in RPMI + B27 medium with 1% Penicillin Streptomycin (Pen-Strep; Thermo Fisher Scientific, Waltham, MA, USA) for 7 days.

### 2.6. Image Analysis

Scaffold images were captured using a digital camera (Canon EOS 6D, ISO 3200, exposure 1/40, image size 5472 × 3648 pixels) attached to a dissection microscope (Olympus SZ40). All images were captured using the same camera settings. Images with a set scale were imported into ImageJ and analyzed manually using the angle and line tools. Three parameters were recorded: the intersection angle of the fibers, fiber diameter, and the spacing between the fibers. Similarly, images of engineered tissues were captured using the same setup and imported into ImageJ. The images were manually analyzed using the area and line tools to quantify tissue compaction, as well as the cross-sectional area for the mechanical analysis.

### 2.7. Mechanical Testing of Engineered Tissues

After 7 days of culture, tissue constructs were excised from the capture frames and attached to a mechanical testing apparatus held in a 37 °C bath of Tyrode’s solution (1.8 mM of CaCl_2_, 1.0 mM of MgCl_2_, 5.4 mM of KCl, 140 mM of NaCl, 0.33 mM of NaH_2_PO_4_, 10 mM of HEPES buffer, 5 mM of glucose). The tissues were allowed to undergo stress relaxation to reach a steady state, then they were paced at 1 Hz for a period of 15 s. Contractile force and displacement were recorded. The cross-sectional area was determined via image analysis and used to calculate the contractile stress of the tissues.

### 2.8. Immunohistochemical Staining and Imaging

The tissues were fixed in 4% paraformaldehyde in PBS and embedded in frozen blocks (Tissue Tek O.C.T., Sakura). The frozen blocks were sectioned at a 10 µm thickness, stained with DAPI (nuclear) and α-actinin (sarcomeric), and imaged (Nikon Eclipse Ti-E Microscope and Olympus FV3000 Confocal Microscope).

### 2.9. Statistical Analysis

Statistics were calculated by using a two-sample t-test and two-way ANOVA, followed by the Tukey–Kramer method of post hoc analysis, with *p*-values of <0.05 being considered statistically significant. All of the analysis was performed in MATLAB and Statistics Toolbox Release 2020 b (MathWorks).

## 3. Results and Discussion

### 3.1. Optimization of PCL Fiber Fabrication

Wet-spinning conditions for the PCL fibers were optimized using a range of concentrations and flow rates. Figure 2A shows a heat map of all tested conditions with a 1–10 scale of fiber rating, based on a subjective rating of fiber smoothness without manual intervention from the operator. Ratings ranged from 1 (rough, uneven, poor quality) to 10 (smooth, uniform, excellent quality). Fibers with lower ratings indicate clumping and inconsistent fiber diameter compared to the smooth and consistent fibers evaluated at higher ratings.

Optimal wet-spinning conditions were determined to be a 20% *w*/*v* solution of PCL in acetone and a 5 mL/h flow rate, which received the highest rating (Figure 2A). These results differ slightly from earlier studies of others, which utilized lower flow rates (0.8–1.75 mL/h) and less concentrated solutions (15–20% *w*/*v*); however, those studies do not report extensive optimization of the wet-spinning parameters [29,30].

The needle diameter was also varied and showed no observable impact on fiber quality. Figure 2B shows fiber diameters produced from 21- and 30-gauge needles at the established flow rate and concentration above. Henceforth, fibers produced from 21-gauge needles are referred to as ‘large fibers’, and fibers produced from 30-gauge needles are referred to as ‘small fibers’.

The fiber diameter was 339 ± 31 µm (*n* = 15, mean ± SD) for large fibers and 105 ± 9 µm for small fibers (*n* = 10, mean ± SD) (Figure 2B). From these results, needle inner diameter is a reliable predictor of resulting fiber diameter, as the 30-gauge needle has an inner diameter of 159 µm and the 21-gauge needle has an inner diameter of 514 µm. The actual fiber diameter produced is smaller, likely due to the stretching of the fiber as it is collected during the solidification process. The resulting fiber diameters range from 59.9 to 72.0% of the needle inner diameter.

The fiber surface was slightly rough, with ridges running longitudinally along the fiber (Figure 2E). The surface and diameter were consistent along the length of the fiber, which is a result of the optimization of the wet-spinning process.

Once the fibers are wet-spun and dried, they can be wound onto the Arduino-controlled mandrel to create scaffolds. Scaffold production has two main quantifiable parameters: fiber angle and pitch. The angles between fibers (Figure 2C) are confirmed as follows: predicted 0° scaffolds had a measured angle of 1.20 ± 0.93°, predicted 30° scaffolds had a measured angle of 27.99 ± 1.77°, predicted 60° scaffolds had a measured angle of 57.52 ± 0.96°, and predicted 90° scaffolds had a measured angle of 90.19 ± 1.65°. The pitch between fibers is defined as the perpendicular distance between the centers of parallel fibers. The pitch was programmed as 400 µm for all groups and resulted in an actual pitch of 481 ± 22 µm (0° scaffolds), 571 ± 40 µm (30° scaffolds), 695 ± 174 µm (60° scaffolds), and 670 ± 107 µm (90° scaffolds).

Small errors due to the step size tolerance of the motors have a minimal impact on the angle of the fibers due to the design of the mandrel. However, these same errors have a larger effect on the pitch between fibers. Thus, we achieve tight tolerances on the fiber angle but have larger variation in fiber pitch, which is acceptable for our chosen application. Given the tight tolerances of the angle between fibers produced by the mandrel, we are able to fabricate consistent scaffolds at specific, user-defined angles. This is useful in adjusting the anisotropic ratio, or ratio of stiffness along a transverse direction to stiffness along the lateral direction of a scaffold.

### 3.2. Single-Fiber Mechanical Analysis and Degradation

To better understand the mechanics of the composite scaffold, we first characterized the mechanical properties of single fibers. Tensile testing (Figure 3A) was used to determine the Young’s modulus within different strain regimes up to 20% strain by binning with 5% strain increments (0–5%, 5–10%, 10–15%, and 15–20%). Young’s modulus for each bin was determined to be in the following ranges: [0–5%: 123–137 MPa, 5–10%: 21–30 MPa, 10–15%: 5.8–6.1 MPa, 15–20%: 0.42–0.63 MPa, *n* = 10]. In a pull-to-break test, the ultimate tensile strength was determined to be 16.1–18.3 MPa (*n* = 5).

The wet-spun PCL fibers exhibit a characteristically long strain-hardening region, reaching over 1200% strain before fracturing, which is consistent with previous studies [31]. Plastic deformation occurs at around 10–15% strain for these fibers in the region where the curve begins to flatten. This higher elastic limit of PCL compared to other scaffold materials is more compatible with the strain experienced in the myocardium, which can be in excess of 20% during healthy function [32].

As PCL is a biodegradable material and selected for both its mechanical properties and degradation, it is important to understand how degradation impacts the mechanics of fibrous scaffolds. The degradation of PCL in vitro and in vivo is facilitated by the hydrolytic degradation of ester groups, which results in random slow chain scission throughout the material when exposed to a salt solution [19]. A longitudinal ANOVA of mechanical data during the course of in vitro degradation revealed a loss of Young’s modulus by up to 50% over a two-week period of degradation in PBS solution at 37 °C (*p* = 0.000022; Figure 3B). Additionally, a two-sample t-test analysis on day 0 and day 14 in salt solution demonstrated a significant decrease in the Young’s modulus of both small and large fibers (*p* = 0.0154 and *p* = 0.0173, *n* = 5; Figure 3C). There was no significant change in mass over the two weeks for either the small or large fibers (*p* = 0.1; Figure 3D), suggesting the scission of polymer chains without a loss of mass.

Significant variability in Young’s modulus between samples (as evidenced by the large standard deviation) is likely caused by inconsistencies in fiber diameter, as stress was calculated using the mean diameters of small and large fibers. Thus, small differences in fiber diameter between samples would have more pronounced effects on cross-sectional area and result in larger variability in stress and Young’s modulus between samples.

These results are encouraging, as they indicate that while there is an overall decrease in Young’s modulus at the two-week time point, the PCL fibers retain sufficient mechanical integrity after two weeks of in vitro degradation to provide mechanical support for tissues as an implantable scaffold material that can be slowly replaced by regenerating host tissue. This is largely consistent with a previous study that analyzed PCL degradation at two- and four-month timepoints, reporting a 57% decrease in Young’s modulus after two months in salt solution [33]. The decrease in Young’s modulus in the present study is 49% ± 29% for small fibers and 35% ± 13% for large fibers after 14 days. It is worth noting that in the aforementioned study, the authors utilized PCL membranes, which would have a larger surface area exposed to solution. Surface area may also account for the difference in Young’s modulus between our small and large fibers. Nonetheless, from these data we can conclude that the mechanical degradation of PCL begins shortly after exposure to a salt solution, giving insight into the short-term degradation of PCL. PCL degradation has been assessed subcutaneously and intravenously in vivo, showing a similar mechanical degradation timeline and mechanism to in vitro studies, indicating that our in vitro analysis of the mechanical properties of degradation is applicable to the in vivo environment [34,35]. In the context of an implantable cardiac patch, this degradation timeline would allow for patch robustness during surgical implantation at two weeks and the mechanical support of native tissues for the first few months, followed by complete degradation of the scaffold in the first 2 to 3 years following implantation.

### 3.3. Acellular PCL Scaffold Mechanical Analysis

Whole scaffold mechanics were analyzed using bare, large fiber scaffolds via a cyclic tensile testing protocol, which was performed at the three different strain rates of 0.1%/s, 1%/s, and 10%/s. The latter is on the same order of magnitude as seen in the native heart wall at 60 bpm (~40% strain/s). Figure 4A shows the stress–strain curves at the different strain rates. There is a noticeable difference in traces for 0° and all angled scaffolds across the strain rates. For all three strain rates, Young’s modulus for 0° scaffolds (63.86 ± 10.08 MPa for 0.1%/s, 99.23 ± 5.69 MPa for 1%/s, and 120.8 ± 1.90 for 10%/s) is significantly higher than the 30° (37.29 ± 4.81 MPa for 0.1%/s, 50.64 ± 9.74 MPa for 1%/s, and 60.34 ± 11.41 for 10%/s) and 90° (36.91 ± 5.15 MPa for 0.1%/s, 43.08 ± 6.42 MPa for 1%/s, and 49.31 ± 6.90 for 10%/s) scaffolds, as well as in the 1%/s and 10%/s strain rates for 0° vs. 60° (53.59 ± 2.46 MPa for 0.1%/s, 64.23 ± 2.90 MPa for 1%/s, and 73.59 ± 3.167 for 10%/s) (Figure 4B). Additionally, for all three strain rates, the area between the loading and unloading curves, or hysteresis, is lower in the 30°, 60°, and 90° scaffolds (Figure 4C).

Measuring the strain-rate dependent properties of the scaffolds allows insight into their viscoelastic behavior in relation to the native heart [36]. The strain rate dependence of the loading and unloading curves is consistent with the viscoelastic properties of PCL that were measured previously [37,38]. The lower Young’s modulus in the angled scaffolds verifies the ability to modulate scaffold mechanics by placing fibers at angles, rather than in a parallel configuration, as predicted by previous studies [6,22]. Interestingly, despite a steep drop-off in Young’s modulus between the 0° and 30° scaffolds, there are no significant differences in Young’s modulus between the 30°, 60°, and 90° scaffolds. This suggests that fiber angles between 30° and 90° may lead to similar longitudinal mechanical properties, but that modulating the angle may still impact the overall anisotropy of the scaffold, in addition to the impacts on tissue compaction in the transverse direction, as discussed previously. Importantly, we have shown that angling fibers results in a significant reduction in stiffness, which is an important result, as it indicates that tissues grown on 30°, 60°, and 90° scaffolds are subject to a different mechanical environment than those grown on 0° scaffolds.

Additionally, the lower hysteresis in the 30°, 60°, and 90° scaffolds suggests that there is less energy dissipation in those structures, which is more consistent with the energy conservation seen in healthy native tissues [39,40] and points to improved mechanical biomimicry in the angled scaffolds. In this way, our scaffolds are customizable not only in terms of mechanical anisotropy, but also in their ability to recapitulate the energy-loss mechanics of native tissue. More broadly, this result demonstrates the potential of our scaffolds to have an active role in supporting native heart tissue to reduce the harmful remodeling that occurs in infarcted regions.

### 3.4. Assessment of hiPSC-CM Tissue Formation, Compaction, and Mechanics on PCL Scaffolds

After extensive characterization of the PCL material and scaffold mechanical properties, the scaffolds were seeded with purified hiPSC-CMs in a collagen hydrogel to form engineered tissues. Five tissue conditions were produced: control tissues supported by posts to provide passive tension (described previously [41]), and tissues seeded on 0° and 30° scaffolds, which, as we have previously shown are comparable to the mechanical anisotropic ratio of the native myocardium [24]. The scaffolds are made of either small or large fibers. In Figure 5C,D, SD and LD represent all tissues with small or large diameter fiber, respectively, regardless of fiber angle design; 0° and 30° represent all tissues with the specified fiber angle design, inclusive of both SD and LD fibers.

A compaction analysis revealed that tissues in all conditions reach a steady, compacted state by day 5 of culture (Appendix A), and that all tissues compact to around 40% of their initial area. Interestingly, compaction on the first day after cell seeding is more pronounced in the tissues grown on PCL scaffolds than in non-scaffold controls (Figure 5C). However, this trend is nullified by day 3, and both the control and scaffolded tissues level out (Figure 5D). When the length and width of the tissues are compared, there is a noticeable difference between the tissue lengths (defined as the larger dimension) of 0° and 30° scaffolds, while there is no difference between their widths (Figure 5B). Interestingly, the angle between the fibers does affect the length of the resulting compacted tissue, with 30° tissues compacting less along the axial direction than their 0° counterparts. This is likely due to the increased transverse resistance that the 30° scaffolds provide, along with an effective ‘cross bracing’ effect of the overlapping fibers to prevent longitudinal motion during compaction, whereas in 0° scaffolds, there is minimal geometric resistance to the axial migration of the cells.

Once compacted, the engineered tissues express an observable, spontaneously beating phenotype (Appendix A). The results from the fluorescent live/dead viability assessment reveal that there is sparse cell death within engineered tissue constructs, which is not unusual and could be due to residual cytotoxic chemicals or routine CM handling (Appendix A). Importantly, there is a significant presence of live cells on and around the fibers. Further, the beating phenotype expressed by the tissues, coupled with compaction data indicating minimal differences between the control and experimental groups, is a strong indication that these tissues are not systematically affected by individual cell death and have sufficient viability for downstream applications.

Engineered tissue morphology was investigated via immunofluorescence staining using DAPI (a nuclear stain) and α-actinin (a cardiac marker of the sarcomeric z-disk). In the control as well as the 0° and 30° small fiber conditions, there was an abundance of α-actinin and DAPI found throughout the tissue constructs (Figure 5G). In the 0° condition, fragments of a PCL fiber fluoresce with the DAPI stain (Figure 5G(iii)). In the 30° condition, the space left behind from two adjacent PCL fibers can be seen (Figure 5G(v)). At 60× magnification, the individual cell morphology shows distinct cardiac sarcomeres (arrows, Figure 5G(ii,iv,vi)). The similar distribution and prevalence of both DAPI and α-actinin in all three constructs indicate that high-level tissue morphology is not adversely impacted by the presence of PCL fibrous scaffolds. Additionally, based on the abundance of sarcomeres, there is good evidence that CMs are distributed throughout the tissues, leading to uniform contractile capability throughout each tissue.

Engineered tissue function was investigated using a mechanical testing apparatus for live tissue, which measured contractile force in hydrated tissues at ~37 °C, while electrically paced at 1 Hz. The metrics used to determine tissue mechanics are peak contractile stress, upstroke velocity, time to 50% relaxation (t_50_), and time to 90% relaxation (t_90_) (for a graphical explanation of each, see Appendix A). Peak contractile stress was significantly higher in the control than in the 0° and 30° conditions (Figure 5F), as was the upstroke velocity (Appendix A). The t_50_ and t_90_ times showed little difference between groups (Appendix A), except between the control and 30° conditions.

While the scaffolded 0° and 30° tissues had smaller contractile forces than the control tissues, various experimental factors may explain why. Firstly, CM contractile force depends on the amount of stretch that the cell experiences through the Frank–Starling Law, which dictates that contractile force increases with increased stretch [42]. Thus, between both differences in attachment (with pre-formed holes in controls vs. punctured holes in scaffolded tissues) and stiffer fibers able to bear the load of stretch (thus, unloading CMs during culture and testing), the CMs may not have experienced the same amount of load and stretch as in the control tissues. Additionally, the lesser compaction seen in small fiber tissues vs. controls (Figure 5D) influences the stress calculation (i.e., normalized per cross sectional area), and previous research indicates that tissue compaction has a profound effect on contractile force [43]. Finally, given the viscoelastic nature of PCL, it may act as a mechanical dampener to absorb energy from CM contractions, as may occur in native viscoelastic tissues. Future studies could incorporate an analysis of fiber motion and strain or optimize fiber number and diameter to further customize this viscoelastic PCL scaffolded engineered tissue for cardiac regeneration.

The customizable scaffold system described herein offers a novel approach to biomechanically reinforcing the infarcted heart and restoring contractile function by cardiomyocyte delivery. By modulating the elements of mechanical anisotropy, energy loss, fiber density, and fiber diameter, the wet-spun fiber-based system gives broad control over multiple scaffolded tissue design considerations. Given this variety of features, our wet-spun fibrous scaffolds offer several advantages over other PCL scaffolds. The most common PCL scaffold type, electro-spun scaffolds, have only recently been shown to be controlled in their anisotropy and fiber diameter [22,44], but lack fine control over fiber density, leading to overly dense scaffolds that prevent cell infiltration into the scaffold [45]. Other scaffolds made from PCL nanofilms [46,47] lack the anisotropic mechanics seen in the native heart and valued for cardiac repair [48]. In contrast, our scaffolds offer finely customizable fiber spacing and angle, which in an in vivo setting would allow for the anisotropic reinforcement of native tissue, while enabling the design of engineered tissue morphology and allowing integration with the host.

## 4. Conclusions

Engineered cardiac tissue patches have the potential to become a revolutionary clinical therapy for the infarcted myocardium through the introduction of new contractile tissue coupled with mechanical reinforcement of the infarct. This work aimed to develop and characterize a customizable scaffold for an epicardial patch application, as well as assess its performance in vitro. Our results show that wet-spun PCL fibers form robust scaffolds using a customizable platform that enables the modulation of scaffold anisotropy while facilitating hiPSC-CM tissue formation. The scaffold production process has been optimized and evaluated to ensure consistency among scaffolds while providing multiple design inputs for the end user. Optimization of the wet-spinning process determined a flow rate of 5 mL/hour of 20% *w*/*v* PCL in acetone and yielded fibers of two unique diameters, 339 ± 31 µm and 105 ± 9 µm, from 21- and 30-gauge needles, respectively. The PCL scaffolds and capture system do not impede tissue formation or compaction, but rather allow for the modulation of fiber orientation, and thus, mechanical cues for tissue development, as well as provide a robust support material for surgical handling. We have empirically verified the compatibility of these wet-spun scaffolds to support contractile engineered hiPSC-CM tissues in vitro. Further, the biodegradation profile of PCL allows for sufficient mechanical support during tissue development and culture but will ultimately leave behind an entirely biological construct absent of PCL in the long-term post-implantation. In conclusion, anisotropic viscoelastic wet-spun PCL scaffolds are well-suited for continued use in engineered cardiac tissues and offer a variety of features to enhance the development and translation of engineered tissues for clinical applications. Future studies should utilize the customization of these scaffolds to produce optimally suited conditions for in vitro tissue development, as well as assess the functional and morphologic impacts of the scaffolds on native tissues when implanted in vivo.

## Figures and Tables

**Figure 1 polymers-14-04571-f001:**
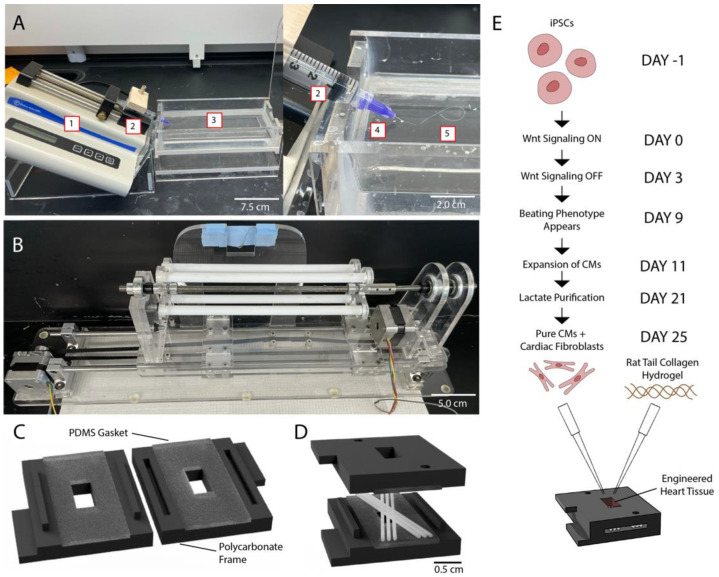
Custom production process allows for facile fabrication of fibrous PCL scaffolds for engineered heart tissue. (**A**) Image of wet-spinning process: a syringe pump {1} extrudes PCL solution {2} into a coagulating ethanol bath {3} through a blunt needle {4} to precipitate a solid PCL fiber {5}. (**B**) Image of Arduino-controlled mandrel. Mandrel rotates and translates to position fibers into a scaffold with user-defined spacing and angle inputs. (**C**) PDMS gaskets are embedded in 3D-printed polycarbonate press-fit capture frames and autoclaved. (**D**) Capture frames are used to remove fibers from the mandrel and prepare scaffolds for tissue culture. (**E**) Differentiation of hiPSC-CMs and formation of engineered tissues: iPSCs are differentiated into CMs using a Wnt signaling pathway, expanded, purified, and combined with a collagen hydrogel to produce engineered heart tissues.

**Figure 2 polymers-14-04571-f002:**
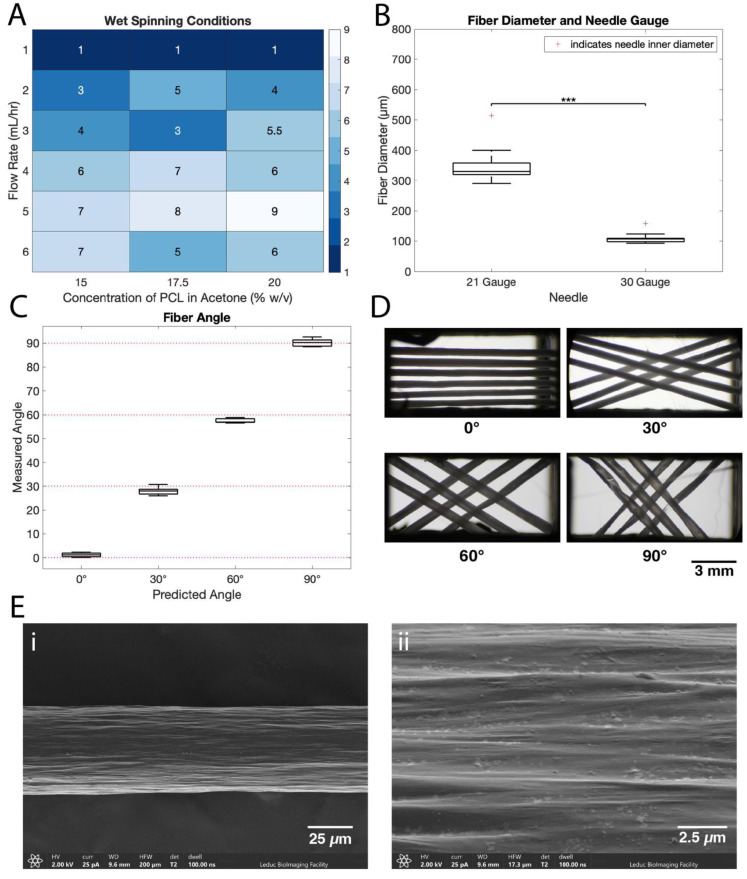
Optimized fiber and scaffold fabrication process produces consistent scaffolds with user-defined parameters. (**A**) Heatmap of fiber quality resulting from varying flow rate and PCL concentrations. Ratings are given by operator observation, ranging from 1 (rough, uneven, poor quality) to 10 (smooth, uniform, excellent quality). (**B**) Boxplot of fiber diameters produced when utilizing two different needle sizes. Needle inner diameter is shown by the red crosses. (*** indicates *p* ≤ 0.001) (**C**) Measured angle and predicted angle between fibers for 0°, 30°, 60°, and 90° scaffolds from image analysis (*n* = 5 scaffolds in each condition). (**D**) Representative images of scaffolds at 0°, 30°, 60°, and 90° fiber orientations. (**E**) SEM images of the surface of a single wet-spun PCL fiber at (i) low and (ii) high magnification.

**Figure 3 polymers-14-04571-f003:**
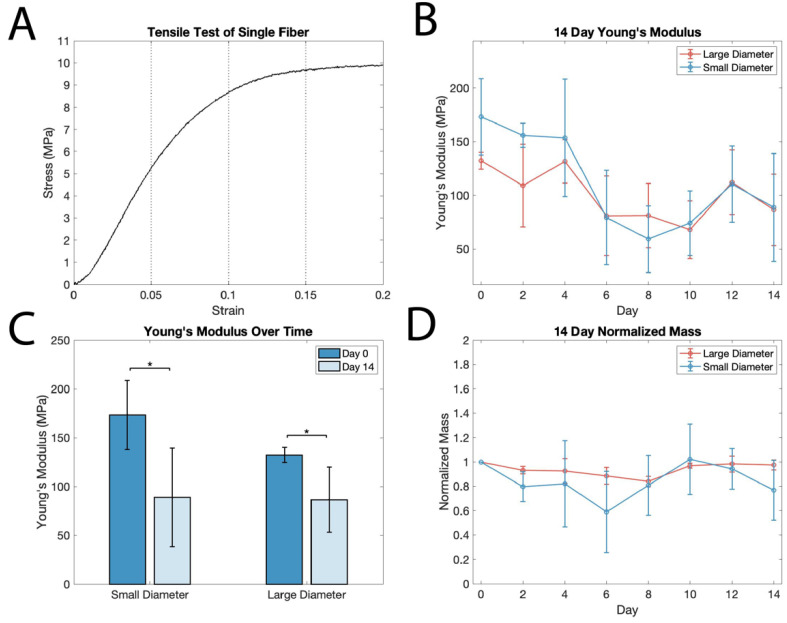
Single-fiber mechanics during degradation show decreases in stiffness but retention of mechanical integrity throughout 2 weeks incubation in salt solution. (**A**) Representative stress–strain curve of a single-fiber tensile test, with bins from 0 to 5%, 5 to 10%, 10 to 15%, and 15 to 20% strain denoted by vertical dotted lines. (**B**) Young’s modulus of small and large fibers at 0–5% strain over a 14-day period in PBS solution (*n* = 5 samples for each time point). (**C**) Young’s modulus of small and large fibers compared at days 0 and 14 (* indicates *p* ≤ 0.05). (**D**) Mass of small and large fibers over 14 days in PBS solution, normalized to initial mass (*n* = 5 samples for each time point; n.s.).

**Figure 4 polymers-14-04571-f004:**
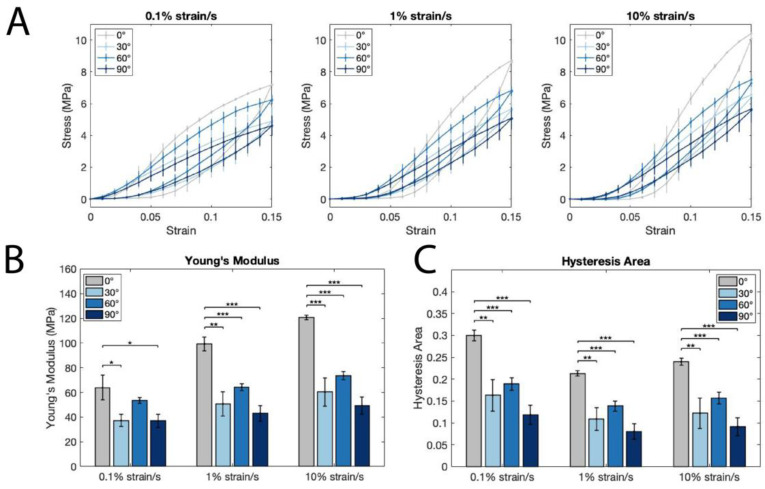
Mechanics of large diameter acellular scaffolds show strain-rate dependence and energy loss lessened with angled vs. parallel fibers. (**A**) Stress–strain curves generated from cyclic tensile testing of scaffolds at strain rates of 0.1, 1, and 10 percent strain/second. (**B**) Young’s modulus of 0°, 30°, 60°, and 90° scaffolds at three different strain rates. (**C**) Hysteresis of 0°, 30°, 60°, and 90° scaffolds, calculated as the area between the loading and unloading curves for the three strain rates (*, **, and *** indicate *p* ≤ 0.05, 0.01, and 0.001, respectively).

**Figure 5 polymers-14-04571-f005:**
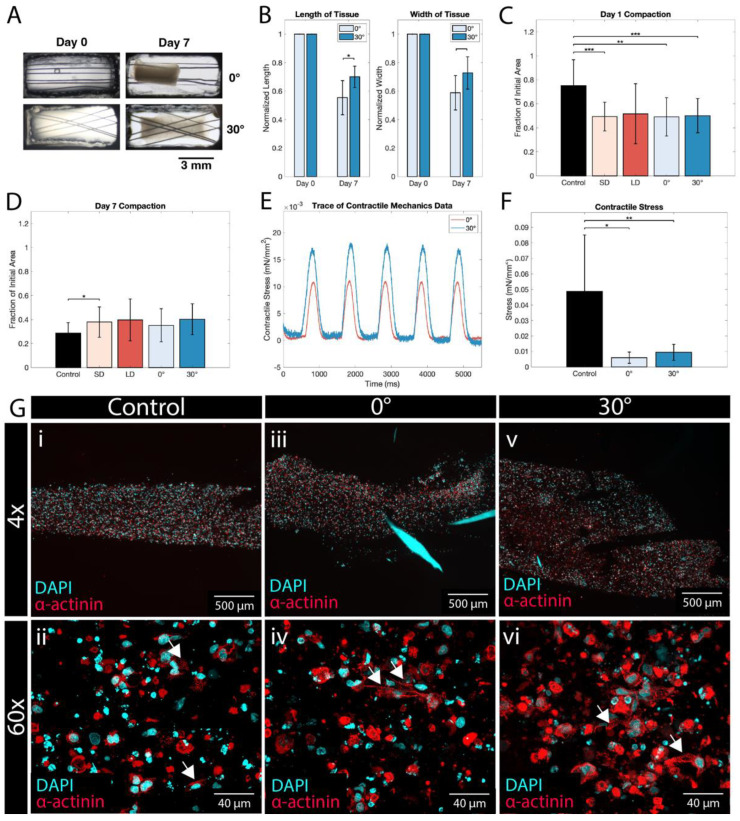
hiPSC-CMs form compacted, contractile tissues on PCL scaffolds. (**A**) Representative images of tissues grown on 0° and 30° scaffolds at day 0 and day 7 time points. (**B**) Length and width of tissues compared at day 0 and day 7 time points. (**C**) Day 1 relative compaction of tissues, normalized to day 0 area (SD: 0° and 30° small diameter tissues, *n* = 6; LD: 0° and 30° large diameter tissues, *n* = 27; 0°: small and large diameter tissues, *n* = 13; 30°, small and large diameter tissues, *n* = 20; control, *n* = 12). (**D**) Day 7 relative compaction of tissues, normalized to day 0 area (SD: 0° and 30° small diameter tissues, *n* = 6; LD: 0° and 30° large diameter tissues, *n* = 27; 0°: small and large diameter tissues, *n* = 13; 30°, small and large diameter tissues, *n* = 20; control, *n* = 12). (**E**) Representative trace of contractile stress generated at 1 Hz pacing by 0° and 30° tissues. (**F**) Contractile stress of control, 0°, and 30° tissues, calculated as contractile force/cross-sectional area. (**G**) Immunofluorescence staining of engineered tissues using DAPI and α-actinin. Widefield fluorescence images of (i) control, (iii) 0°, and (v) 30° tissues at 4× magnification. Confocal microscopy images of (ii) control, (iv) 0°, and (vi) 30° tissues at 60× magnification; cardiac sarcomeres are highlighted by white arrows. The large areas of cyan fluorescence seen in (iii) are fragments of PCL fibers, which have fluorescent properties. *, **, and *** indicate *p* ≤ 0.05, 0.01, and 0.001, respectively.

## Data Availability

Datasets generated during this study will be made available upon request to the authors.

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
