# Peer review of "Wet-Spun Polycaprolactone Scaffolds Provide Customizable Anisotropic Viscoelastic Mechanics for Engineered Cardiac Tissues"

_polymers, 2022, doi:10.3390/polym14214571_

Round 1

Reviewer 1 Report

The manuscript “Wet-Spun Polycaprolactone Scaffolds Provide Customizable Anisotropic Viscoelastic Mechanics for Engineered Cardiac Tissues” by Schmitt et al. Is a well written article the microfiber scaffold that can have manipulated mechanical anisotropy. These fibers are also modified to hydrogels to form composite engineered cardiac tissue. The following points can be addressed 

  1. 1. Provide some SEM and TEM images of the wet-spun PCL microfibers and hiPSC-CMs in a collagen  hydrogel 

Reviewer 2 Report

This is a timely effort by authors on "Wet-Spun Polycaprolactone Scaffolds Provide Customizable Anisotropic Viscoelastic Mechanics for Engineered Cardiac Tissues". However, there are few addressable observations from my side:

1. Novelty needs to be clarified in a more precise way.

2. Are all the methods of testing mechanical integrity of scaffolds standardized?

3. have you used the standard procedure for scaffold and engineering tissue preparation? if yes, then there is no point of mentioning all the procedure which will increase the unnecessary length of paper. This approach can be adopted for all the tests and methodologies throughout this manuscript.

4. What were the operational setting of digital camera for image analysis? and was it standardized too?

5. Conclusion usually does not contain any literacture citations.

6. You should clearly write the optimization result of wet spun process in the conclusion section as you somehow highlighted in the abstract.
